# Latest Findings in Immunoglobulin Y Technologies and Applications

**DOI:** 10.3390/ijms26136380

**Published:** 2025-07-02

**Authors:** Robert Capotă, Dana Ciaușu-Sliwa, Andra-Cristina Bostănaru-Iliescu, Valentin Năstasă, Mihai Mareș

**Affiliations:** Faculty of Veterinary Medicine, “Ion Ionescu de la Brad” Iasi University of Life Sciences, 700490 Iasi, Romania; robert.capota@iuls.ro (R.C.); dana.ciausu@iuls.ro (D.C.-S.); andra.iliescu@iuls.ro (A.-C.B.-I.); valentin.nastasa@iuls.ro (V.N.)

**Keywords:** immunoglobulin Y, hyperimmune egg, One Health, antimicrobial resistance

## Abstract

Immunoglobulin Y (IgY), the major antibody class in birds, has gained increasing attention in recent years as a versatile and ethically sustainable alternative to mammalian immunoglobulins. IgY has demonstrated strong potential in diagnostics, prophylaxis, and therapy across a wide range of fields, including infectious diseases, allergy management, oral health, and food safety. Its applications in animal health—particularly in poultry, livestock, and companion animals—further underscore its relevance within the One Health framework. This review provides a comprehensive synthesis of IgY technology, starting with its physiological role in maternal immunity and the structural characteristics that distinguish it from mammalian immunoglobulin G (IgG). This review outlines current strategies for IgY production and purification. It also provides an overview of its biomedical and veterinary applications, including its use in diagnostics, prevention, and treatment—such as for SARS-CoV-2—primarily based on studies published in the past five years. The final section addresses the current limitations of IgY technology, such as variability in protocols, stability challenges, and the need for safety assessment, while highlighting the importance of harmonized guidelines to support broader implementation. With growing scientific interest, expanding clinical research, and increasing availability of commercial products, IgY is well positioned to become a valuable immunobiological tool for both human and veterinary applications.

## 1. Introduction

Maternal immunity refers to the transfer of protective antibodies from the mother to her offspring, providing temporary immune defense during the early stages of life when the immune system of the neonate is not yet fully developed [1]. In mammals, this transfer occurs either transplacentally before birth or through the intestinal absorption of immunoglobulin G from colostrum within the first 24 h after birth [2]. In birds, however, the process involves immunoglobulin Y (IgY), the predominant antibody class, which also plays a central role in the passive transfer of immunity. This transfer occurs via the egg yolk through a specialized, two-step mechanism: first, IgY is transported from the maternal bloodstream into the yolk; then, during embryonic development, it is absorbed from the yolk into the embryo’s circulation via the yolk sac membrane [3].

In recent years, IgY has gained prominence as a versatile tool in biomedical research and applications, due to its phylogenetic distance from mammals, which enhances the immune response to conserved mammalian proteins. Moreover, hens can be immunized with specific antigens to generate large quantities of high-affinity antibodies, which are then non-invasively harvested from egg yolks—yielding between 1000 and 2800 mg of IgY per month per hen, a volume comparable to that obtained from larger mammals such as sheep or goats [4]. This production system supports the ethical principles of the 3Rs (Replacement, Reduction, and Refinement) while also offering cost-efficiency and scalability. The resulting antibodies demonstrate remarkable thermal and pH stability, which enhances their usability in diverse formulations for oral, topical, or intranasal delivery.

In addition to its diagnostic potential, IgY has shown efficacy in treating or preventing infectious diseases in both human and veterinary contexts. It has been successfully applied against pathogens such as *Helicobacter pylori*, *Pseudomonas aeruginosa*, *Clostridioides difficile*, and SARS-CoV-2, with minimal side effects and no reported risk of inducing resistance. The versatility of IgY positions it as an attractive alternative to antibiotics, particularly in the context of growing antimicrobial resistance.

This article presents a detailed analysis of IgY, beginning with its physiological and structural characteristics, followed by a brief overview of current production technologies. It then reviews the latest biomedical applications of IgY, focusing on findings from recent studies, including clinical trials. The synthesis concludes by discussing the current advantages, limitations, and practical considerations of IgY technology, emphasizing its potential for broader implementation in both research and applied immunotherapy.

## 2. Structure and Function of IgY

Immunoglobulin Y is a class of antibodies found in birds, reptiles, amphibians, and lungfish [4]. IgY is similar to IgG found in mammals, and it was called IgG until it was discovered that distinct differences exist between the two, prompting the switch to the term ‘IgY’ [5,6].

The structure of immunoglobulin Y is comparable to mammalian IgG, with their basic unit structure consisting of two light and two heavy polypeptide chains that form a monomeric unit (Figure 1). The two heavy chains are connected to each other via disulfide bonds, while each heavy chain is attached to a light chain through a disulfide bond as well. In any immunoglobulin molecule, the two heavy chains and two light chains are identical, resulting in an antibody with two identical antigen-binding sites. This symmetry enables the antibody to bind simultaneously to two identical structures [7]. However, while IgG has three constant domains per each heavy chain, IgY is comprised of four constant domains per heavy chain, an aspect that accounts for the higher molecular weight of IgY (180 kDa for IgY compared to 150 kDa for IgG) [8,9]. The extra domain of the heavy chain of IgY can also play a significant role in providing more molecular stability as compared to IgG [10].

Another important structural difference of IgY compared to IgG is the absence of a genetically encoded hinge, which results in limited flexibility. This has implications in its biological properties, such as its inability to precipitate antigens at physiological salt concentrations [11], a phenomenon observed in both chickens [12] and ducks [13]. One notable advantage of the higher chain stiffness of IgY is its potential association with increased resistance to proteolytic degradation and fragmentation, as reported in previous studies [14].

The Y-shaped structure comprises two main regions; the fragment-crystallizable (Fc) and the fragment-antigen-binding (Fab) region [15]. Similar to IgG, the Fc region of IgY mediates most of its biological effector functions, including complement fixation and opsonization [16]. Cell surface Fc receptors interact with the constant (Fc) region of the immunoglobulin, contributing significantly to the regulation and modulation of the immune response [17]. The Fab (fragment antigen-binding) region of an immunoglobulin is primarily responsible for recognizing and binding to specific antigens [18].

The Fc portion is unable to activate the human complement system [19], a property that has important implications for ELISA and other clinical immunoassays. By avoiding complement activation, IgY minimizes the background interference caused by varying complement activity in patient samples, thereby enhancing analytical accuracy, reducing false positives, and improving overall assay reproducibility [16]. Immunoglobulin Y does not bind to the rheumatoid factor, nor does it activate human anti-mouse antibodies [20], furthermore proving its potential to reduce interference problems in immunological assays [16]. In addition, chickens generate a stronger immune response against conserved mammalian proteins due to their significant phylogenetic distance from mammals [21]. This phylogenetic separation enables the avian immune system to recognize epitopes that might be weakly immunogenic or even tolerated in mammalian hosts. As a result, IgY antibodies can target unique and conserved antigenic regions with high specificity, making them especially valuable in diagnostics where cross-reactivity must be minimized and the detection of low-abundance or structurally conserved targets is critical [22].

In humans and mice, immunoglobulin (Ig) genes are formed through a random assortment of variable (V), diversity (D), and joining (J) segments, generating a diverse repertoire of antibodies [23]. After antigen stimulation, this diversity is further expanded by somatic hypermutation, and the Ig isotype can be altered through class switch recombination. In chickens, however, the contribution of V(D)J recombination to immunogenetic diversity is minimal as only a single locus undergoes rearrangement to produce a functional gene [4]. Instead, chickens rely on gene conversion to achieve antibody diversity. During this process, short sequences in the V-region gene are replaced with homologous sequences donated by pseudogenes, significantly enhancing the diversity of IgY and compensating for the limited role of V(D)J rearrangement [17].

IgY exhibits remarkable stability across a range of conditions: it remains stable at temperatures between 4 °C and 60 °C and within a pH range of 4.0 to 10.0. Additionally, IgY maintained its stability for 35 days at 4, 25, and 37 °C [24]. Also, eggs stored for up to one month at room temperature, or for up to 6 months at +4 °C, showed no significant reduction in antibody titers [25]. The stability of IgY under heat stress (75–80 °C), acidic conditions (Ph 3), or high pressure (5000 kg/cm^2^) was found to be increased by the addition of high concentrations of sucrose [26].

## 3. Immunoglobulin Y Production and Extraction

Understanding the production and extraction process of IgY is essential for exploiting its full potential in biomedical applications. The technology bridges immunology, biotechnology, and translational research and involves a series of coordinated steps, from antigen selection and hen immunization to antibody extraction and purification. Each stage can influence the quality, specificity, and usability of the final commercial product. As IgY continues to gain recognition as a noteworthy tool for prophylactic, therapeutic, and diagnostic purposes, it becomes increasingly important to ensure not only efficient production but also rigorous testing. Evaluating its interactions with target antigens and the host is critical to confirming safety and efficacy for downstream applications (Figure 2).

### 3.1. Optimizing IgY Production

Optimizing the immunization process remains a critical challenge regarding IgY technology. Key challenges include selecting the appropriate antigen type, choosing the best adjuvant, and designing an effective immunization protocol to maximize IgY yield and specificity. These aspects came into increased attention during 1990, leading to the course of workshops organized by the European Centre for the Validation of Alternative Methods (ECVAM) and the subsequent report [6] (Table 1). This initiative aligned with the ethical framework of the 3Rs—Replacement, Reduction, and Refinement—aimed at minimizing animal use and improving welfare in antibody production. Since then, various protocols have been published and utilized in IgY studies [9,27], emphasizing the need for close monitoring of antibody titers throughout the immunization period to enable adjustments to the protocol based on the observed immune response [17].

The first step in the obtaining of IgY is the choice of an antigen to be used for immunizing hens. Selecting an antigen of the right type and of high quality is paramount for inducing a strong humoral response and, subsequently, the production of IgY antibodies. Immunogenicity, which refers to the ability to stimulate cellular and humoral immune responses, and antigenicity, the capacity to be specifically recognized by the antibodies produced during the immune response, are two essential antigen characteristics for generating high-performance antibodies [17,28].

Various antigen types can be used to produce specific IgY in birds: complex antigens from whole viruses, bacteria, parasites, or fungi [29,30,31] and recombinant proteins [32,33,34,35] or other antigens with immunogenic properties, such as toxoids, venom, allergenic antigens, and even other antibodies [36,37,38,39]. More recent technologies include the use of haptens, small molecules that are converted to antigens by linking to a carrier protein, virus-like particles, bacterial ghosts, virosomes, or nucleic acid vaccines [17,27].

Particularly when working with pathogenic or toxic antigens, a crucial step is antigen inactivation. Various methods can be employed, with the overall aim of preserving immunogenicity and antigenicity while at the same time protecting animal welfare and ensuring the safety of the laboratory team [17]. Various methods can be employed to inactivate antigens, such as chemical inactivation using formaldehyde [40,41,42], physical inactivation through heat [43] or ultrasonication [44], or by combining different types of methods [45].

The antigen dose is another key aspect for IgY production as insufficient doses may fail to elicit an adequate immune response while excessively high doses can overstimulate or overwhelm the immune system, potentially leading to immune tolerance or reduced antibody production. Early studies showed that 0.1 or 0.5 mg of mouse-IgG elicit a maximum response regarding the antibody titer, while increasing to 1 mg can lead to a reduction in the immune response [27]. However, the majority of studies use between 50 and 200 µg of antigen per inoculation [32,33,34,46]. A more recent study further supports the importance of antigen dose in IgY production, demonstrating that while doses as low as 2 µg can elicit significant antibody levels after multiple immunizations, the highest antibody titers were observed with doses of 20 and 200 µg [47].

Other important factors regarding the production of antibodies include the genetic background [48], nutrition [49], age [50], keeping conditions [51], or biorhythm [52].

Choosing the suitable adjuvant also plays a significant role regarding the outcome of the immunization since the inactivated antigens used in the inoculum are usually less immunogenic compared to the modified live microbes [53]. Adjuvants enhance IgY production by improving antigen retention, stimulating immune responses, and increasing specific antibody titers. To produce IgY antibodies in birds, the most frequently used adjuvants are complete and incomplete Freund’s adjuvants (FCA and FIA) [54]. FCA, which is a mixture of a mineral oil and heat-killed and dried mycobacteria, has been the gold standard for generating high levels of antibodies in animals [55]; however, its main disadvantage is the potential to induce local and systemic side-effects [27]. FIA does not contain mycobacteria, which usually is responsible for the adverse reactions seen with FCA [55]; therefore, numerous studies now use FCA for the first immunization and FIA for the booster injections [41,56,57]. Other commercial adjuvants can be used, such as Montanide, Polygen, or aluminum hydroxide [34,47,58,59], while there is still research being done in order to obtain a better, safer adjuvant [60,61].

Usually, antigen administration for IgY production in chicken is done through the intramuscular route, most common into the breast muscle as administration into the thigh muscle may lead to lameness [4]. The second most common route is the subcutaneous route [33,58,62], which may be associated with higher yields of antibodies [27] but may also be less suited for young animals [9]. The intravenous route should only be used without adjuvants and performed very slowly in order to avoid an anaphylactic reaction [17], while the oral route has also been explored as a non-invasive alternative [63] and may be useful for large scale immunizations, although the resulting titers are generally lower than those achieved by intramuscular injection [27,64].

Immunization should be performed when the animals are of egg-laying age, with the overall aim being to make the peak of laying and the peak of antibody production to occur at the same time [65]. Although the total IgY content of an egg has been reported to increase with age from 32 to 52 week-old breeders [66], a recent study showed that the content of IgY from spent laying hens (age 83 weeks) was lower than that isolated from new laying hens (age 27 weeks) [50]. The interval between multiple immunization varies greatly between different studies, with early studies considering a primary vaccination and a booster given before the laying period, at 4–6 weeks interval [6], or every 3–5 weeks [67]. The usual interval is now at least 4 weeks to allow for the generating of immune memory [17], while a general rule is to administer a booster immunization whenever the titer of IgY reaches a plateau or begins to decrease [65]. Other studies reported administering the booster injection at 7 days [62], 8 days [57], 10 days [42], or 14 days [40,41,68]. After the first immunization, IgY begins to appear in the serum in approximately 4 days, while a delay of 3–4 days was observed between the occurrence of IgY in the serum and its occurrence into the egg yolk [17]. High titers of IgY can be kept for more than 150 days through booster immunizations [65].

### 3.2. Extraction and Purification of IgY

To get a better understanding of the technology of IgY production, it is important to discuss the structure of egg yolk. The egg yolk contains numerous nutrients and preservative substances, highlighting its role as an embryonic chamber [69]. The composition of yolk is approximately 48% water, 32% lipid, and 16% proteins [70], and its main components include phosvitin, low-density lipoprotein (LDL), high-density lipoprotein (HDL), and lecithin [71].

From a structural point of view, the egg yolk has two parts: an aqueous plasma consisting of 86% low-density lipoproteins and the remainder α-livetin (serum albumin), β-livetin (α2-glycoprotein), and γ-livetin (IgY) and granules of different sizes composed of 70% high-density lipoproteins (α- and β-lipovitellins), 16% phosvitin (glycophospoprotein), and 12% low-density lipoproteins (LDL) [17]. Since IgY is part of the aqueous plasma, the first step of IgY extraction consists of removing lipids to obtain the water-soluble components (water-soluble filtrate—WSF), while the second step involves the precipitation of antibodies present in the WSF by different techniques [9].

The usual de-lipidation technique is the water dilution method, which results in the aggregation of yolk lipids at low ionic strength [72]. The degree of dilution varies among different authors and ranges between 6 and 10-fold dilution [9], while variations of the technique include pH adjustment of water around 5–5.2 [73] or adding a freeze–thaw cycle for the diluted egg yolk [74]. At the end of the process, the mixture is centrifugated, and the WSF is collected in the supernatant. The water dilution method has the advantage that no toxic products are used, making it suitable for the purification of IgY for oral administration [17,34,75]. Other methods of de-lipidation include the use of polyethylene glycol [76], anionic polysaccharides such as pectin or λ-carrageenan [60,77,78], organic solvents such as chloroform [79], or specific chemicals such as caprylic acid [80].

Four main types of methods can be used to purify the contaminants and concentrate the IgY from the delipidated plasma: precipitation, chromatography, filtration, and aqueous biphasic systems [17]. Salt precipitation is commonly employed in protein purification. The primary mechanism is that most proteins require at least a small concentration of salt to remain stable. At very low salt concentrations, proteins with charged regions tend to aggregate. Salt ions help neutralize surface charges, preventing aggregation, while at high concentrations, excessive charge at the surface of proteins can induce aggregation again [81]. Different saturated salt solutions can be used, such as ammonium sulphate [31], sodium sulphate [82], polyethylene glycol [83], or sodium chloride [34].

Chromatographic methods are generally employed in industrial settings to obtain samples of IgY with the highest purity, due to their high resolution, selectivity, and efficiency [17]. Different methods can be used, such as affinity chromatography [37], ion exchange chromatography [84], or hydrophobic interaction chromatography [85,86]. Although chromatographic methods have proven advantages as mentioned, they are generally expensive and, therefore, impractical for large-scale use [9].

Filtration methods, such as funnel filtration, column-filtration, or ultrafiltration, offer another possibility for IgY purification, with the main principle being that the solution is passed through a barrier that retains some species based on their size and shape while allowing other species to pass through [17]. Column and ultrafiltration are feasible for large-scale use and industrial applications [72].

Aqueous biphasic systems represent a more recent alternative to conventional IgY purification methods, with research showing that these systems maintain the structural integrity of IgY, offer satisfactory levels of purity, and represent a cost-effective, simple, and scalable technique for IgY purification [87].

At the same time, numerous studies employ a combination of different techniques in their IgY extraction protocol, usually salt precipitation by polyethylene glycol or ammonium sulphate followed by chromatographical methods [32,37,88] or ultrafiltration [42], or a combination of all three types of methods [57].

### 3.3. Evaluation of IgY

Antibodies can be characterized by various methods to determine their purity, quantity, and biological activity. These parameters are evaluated independently, and confirmation through a single analytical method is usually sufficient for polyclonal antibodies. There is a close relationship between purity and quantity as each additional step in the extraction methodology leads to a loss of protein content [17].

The determination of protein concentration in IgY extracts is typically performed prior to detailed antibody evaluation. The total protein concentration in IgY extracts is primarily determined using the Bradford method [37,46,89], the Lowry method [83], or the bicinchoninic acid (BCA) assay [68]. In addition, protein concentration can be assessed by ultraviolet absorption at 280 nm, according to the Lambert–Beer law [50,90].

Since IgY has a known molecular weight of 180 kDa, its purity can be evaluated by SDS-PAGE (sodium dodecyl sulfate polyacrylamide gel electrophoresis), considered the gold standard technique and widely used to assess the purity of IgY isolated from egg yolk [9]. Western blotting is a method commonly used in conjunction with SDS-PAGE to confirm the presence and evaluate the purity and specificity of IgY isolated from immunized egg yolks [42,62,91]. For instance, one study [42] employed Western blot to confirm that anti-*Porphyromonas gingivalis* IgY specifically recognized multiple protein bands from strains W83 and ATCC 33277, without cross-reactivity to *Staphylococcus aureus*. High-performance liquid chromatography (HPLC) can also be employed as a complementary technique to further assess purity and identify protein composition with high resolution [60].

### 3.4. Routes of Administration and Pharmacological Considerations for IgY Products

The route of administration is a critical factor in determining the efficacy, safety, and practicality of immunoglobulin Y (IgY)-based interventions, both for therapeutic and prophylactic purposes. Due to their structural and functional differences from mammalian antibodies, IgYs display unique pharmacokinetic (PK) and pharmacodynamic (PD) characteristics that significantly influence their in vivo performance, particularly at mucosal surfaces.

Pharmacokinetically, systemic absorption of IgY is limited due to its large molecular size and the absence of binding to human Fc receptors, which are typically involved in active immunoglobulin transport across biological membranes [92,93]. This contributes to the low systemic penetration observed after parenteral administration. For example, in a porcine pneumonia model, intravenous administration of IgY led to low alveolar concentrations, likely due to the lack of Fc-mediated transport [94]. In the context of mucosal administration, a human study involving intranasal delivery of IgY reported no evidence of a systemic inflammatory immune response [95], reinforcing its reputation as one of the safest therapeutic agents available [93]. When administered orally, IgY exhibits variable stability along the gastrointestinal tract. While it resists degradation by trypsin and chymotrypsin, it is sensitive to pepsin, with degradation depending on pH and the enzyme/substrate ratio [96]. In neonatal piglets, the reported serum half-life of IgY is approximately 1.85 days, while in the gastrointestinal tract, it averages 1.73 h [97]. In humans, biologically active IgY can be detected in saliva up to 12 h after oral gargling [98].

In terms of pharmacodynamics, unlike mammalian IgG, IgY does not exert systemic immunomodulatory effects. Instead, it acts locally at mucosal surfaces, where it binds specifically to pathogens and neutralizes them without causing inflammation or immune-mediated damage. This mode of action makes IgY antibodies well suited for mucosal passive immunization, providing effective local protection at respiratory, oropharyngeal, and gastrointestinal surfaces [93]. For instance, intranasal administration of anti-SARS-CoV-2 IgY for 14 days demonstrated excellent local safety and tolerability in humans [95]. This route has also been explored in animal models, including mice [90,99,100] and hamsters [101,102], further supporting the safety and feasibility of intranasal delivery. Oral administration has likewise been evaluated in various settings. In humans, anti-*Helicobacter pylori* IgY given three times daily for two weeks significantly improved clinical symptoms and quality of life, with a 30.6% increase in bacterial eradication rates [103]. In veterinary applications, oral IgY has been studied in calves [30,104], broiler chickens [105,106,107], and domestic cats. In the latter, a six-month safety evaluation of cat food supplemented with anti-Fel d 1 IgY found no increase in mutagenicity or chromosomal aberrations [108].

## 4. Biomedical Applications of IgY

The unique immunological properties of immunoglobulin Y (IgY)—including high antigen specificity, lack of interaction with mammalian Fc receptors, and minimal background reactivity—make it a valuable tool for diagnostic applications. As outlined in a recent review, IgY can be developed and evaluated through established in vitro methods for the detection of a wide range of biomolecules [9]. Recent studies have demonstrated its effectiveness across various diagnostic platforms, addressing both infectious and non-infectious disease targets (Figure 3).

### 4.1. Avian Antibodies in Diagnostic

Although the primary focus of the reviewed studies is therapeutic and prophylactic, diagnostic applications of IgY are emerging, particularly in research and tool development. CRISPR-Cas systems are increasingly used in diagnostics to detect nucleic acids with high specificity, and the ability to monitor the presence or activity of Cas proteins is critical for quality control, validation, and troubleshooting in CRISPR-based assays. Anti-SpCas9 IgY antibodies have been developed for the immunodetection of CRISPR-Cas components in biological samples, demonstrating high sensitivity and specificity in *Leishmania braziliensis* promastigotes expressing SpCas9 [46]. Such tools are essential in molecular biology for tracking gene editing processes and validating CRISPR efficacy.

Immunoglobulin Y has also been employed to target bacterial pathogens. Anti-SpA IgY antibodies against Staphylococcus aureus demonstrated high specificity without cross-reactivity, reliable performance across various sample types, and advantages over IgG-based assays in terms of reduced false positives. Moreover, the same antibodies exhibited bacteriostatic activity and reduced biofilm formation, highlighting both diagnostic and neutralizing functions [109]. Since biofilms contribute significantly to antimicrobial resistance by shielding bacteria from antibiotics and immune responses, this effect underlines the potential of IgY as a valuable adjunct strategy in antimicrobial resistance control. Similarly, IgY antibodies targeting the LTB subunit of the heat-labile enterotoxin produced by enterotoxigenic Escherichia coli (ETEC) were used to develop a sandwich ELISA, achieving an analytical sensitivity of 39 ng, consistent with previously reported detection ranges [110]. IgY-based assays have also been developed for the rapid detection and discrimination of toxigenic *Vibrio cholerae*, targeting outer membrane protein W (OmpW) and cytotoxin B (CtxB) antigens [111].

In oncology, anti-human B7-H4 IgY antibodies were successfully developed and used to establish a double-antibody sandwich ELISA capable of detecting soluble B7-H4 in human serum with higher sensitivity than a commercial kit [91]. In the context of viral infections, multiple studies have confirmed the potential of IgY-based platforms for the detection of SARS-CoV-2. These include an electrochemical immunosensor utilizing anti-spike IgY with a detection limit of 5.65 pg/mL [88] and an IgY-scFv construct with high affinity for the S1 subunit of the spike protein, capable of recognizing multiple epitopes and showing promise for both diagnostic and therapeutic applications [112]. Additionally, neutralizing IgY antibodies against human papillomavirus (HPV) have been characterized using pseudovirus assays, suggesting their potential utility in neutralization-based diagnostic platforms to assess viral infectivity and vaccine responses [113].

For parasitic infections, IgY antibodies have shown strong diagnostic potential. Anti-*Ascaris suum* IgY achieved high avidity and recognized multiple antigenic bands, with immunofluorescence and immune complex detection methods yielding sensitivity and specificity rates of 80% and 90%, respectively [114]. Similarly, IgY antibodies against Toxoplasma gondii were successfully conjugated with latex particles for the development of a particle-based detection platform, with potential applications across health, food, and biotechnology sectors [115,116]. A coproantigen sandwich ELISA test was also developed using recombinant *Opisthorchis viverrini* cathepsin F (rOv-CF) for the diagnostic of *O. viverrini* infections, the conventional stool examination making the diagnostic difficult given the low intensity of the infection and intestinal coinfections, in the context of fatal cholangiocarcinoma caused by chronic and repeated infections with liver flukes [117].

Beyond human medicine, IgY-based diagnostic tools have shown promising results in veterinary and animal health applications. In canine diagnostics, anti-DEA1.1 IgY antibodies were successfully produced and purified from egg yolk for use in blood typing. The resulting test is simpler to standardize than traditional canine blood group assays, and may facilitate routine screening in clinical practice [118]. In poultry, a lateral flow assay employing anti-IBDV IgY enabled rapid detection of infectious bursal disease virus (IBDV), demonstrating strong specificity by avoiding cross-reactivity with avian influenza and Newcastle disease viruses [119].

In aquaculture, IgY has been utilized for the diagnosis of hemorrhagic septicemia caused by *Aeromonas hydrophila* in *Piaractus mesopotamicus*. Immunohistochemical analysis confirmed the presence of *A. hydrophila* within host phagocytes and surrounding tissues, supporting the applicability of IgY in aquatic diagnostics [120]. Additionally, anti-vitellogenin IgY was developed for detecting reproductive biomarkers in *Cichlasoma festae*, with ELISA assays successfully quantifying vitellogenin concentrations between 10–1280 ng/mL [57].

In the area of food safety, IgY antibodies have been applied for the detection of contaminants and allergens. An indirect competitive ELISA (ic-ELISA) using anti-zearalenone (ZEN) IgY was developed to detect ZEN in post-fermented tea, showing strong correlation with HPLC results and demonstrating potential for rapid screening of mycotoxins in the beverage industry [31]. Similarly, an IgY-based immunosensor targeting dust mite protein was successfully used for the detection of allergens in flour [121].

In research settings, IgY has been used to detect and isolate minor bioactive components. For example, affinity measurements between lactoferrin and anti-lactoferrin IgY revealed that IgY is well suited for immunochemical assays and chromatographic applications targeting low-abundance molecules [122]. Furthermore, conjugated anti-mouse IgG IgY has been proven as a reliable secondary antibody in ELISA, Western blot, and immunofluorescence assays. These conjugates displayed high thermal and pH stability and performed comparably to conventional goat anti-mouse IgG, supporting their use in broad immunological workflows [123].

All these findings underscore the versatility of IgY in biomedical research and suggest a promising future for its diagnostic deployment, especially where mammalian antibodies might cause cross-reactivity or ethical constraints.

### 4.2. The Role of IgY in Prevention and Therapy

The use of IgY antibodies as a passive immunization strategy has shown promising results in combating viral respiratory infections, particularly due to SARS-CoV-2 and MERS-CoV. In vivo studies have demonstrated that intranasal administration of anti-SARS-CoV-2 IgY significantly reduces viral load, lung inflammation, and histopathological damage in hamster models [101,102]. In these studies, IgY was delivered as an intranasal spray (40 μL per dose or 10 μL/nostril) either prophylactically—1 to 2 h before viral challenge—or therapeutically at multiple timepoints post-infection. Treatment regimens included up to three daily doses in the first 48 h, followed by maintenance dosing every 8–12 h until day 4 post-infection. This delivery approach also inhibited spike-mediated cell–cell fusion and ensured antibody persistence in the nasal and oral cavities for several hours after administration [99]. Radiolabeled IgY exhibited preferential accumulation in the trachea and no adverse effects in mice and rats, supporting the safety of mucosal delivery [95,100]. Additionally, IgY antibodies produced against inactivated SARS-CoV-2 demonstrated potent neutralizing activity and suitability for large-scale production [99]. In the case of MERS-CoV, anti-S1 IgY significantly reduced viral antigen expression and lung inflammation in infected mice, further confirming its therapeutic potential against emerging coronaviruses [124].

IgY antibodies have also demonstrated prophylactic and therapeutic potential against a variety of other viral pathogens. In a murine model, intranasal administration of IgY specific to H5N1 influenza virus protected mice from weight loss, lung pathology, and clinical symptoms following infection, without impairing the development of adaptive immunity or immunological memory [90]. Against norovirus, anti-HuNoV IgY effectively disrupted virus binding and replication in a human intestinal enteroid model, showing promise for oral prophylaxis, particularly in populations lacking access to effective vaccines [68]. In the case of dengue virus, specific IgY antibodies neutralized viral infection in vitro, while one variant also conferred significant protection in mice against a lethal challenge [85]. Rotavirus-specific IgY conferred partial protection in cynomolgus monkeys and effectively neutralized viral activity in cell cultures, reducing viral loads in vitro [125,126]. IgY antibodies raised against multiple strains of human papillomavirus (HPV) achieved neutralizing titers of up to 1:2000, suggesting their applicability as therapeutic agents, particularly post-infection [113]. Additionally, in a murine pseudovirus challenge model, high-dose IgY antibodies against Ebola virus provided post-exposure protection and displayed excellent thermostability—an advantage for deployment in resource-limited settings [127].

Regarding antibacterial activity, IgY antibodies targeting *Escherichia coli* and *Helicobacter pylori* have shown strong potential as alternative or adjunctive agents in both food safety and therapeutic applications. In the context of functional food development, anti-*E. coli* O157:H7 and anti-*H. pylori* IgY were successfully incorporated into fortified ice cream, with 0.6 mg/mL being sufficient for antimicrobial effectiveness, suggesting a novel delivery vehicle for gastrointestinal prophylaxis [89]. In a murine model, oral administration of anti-urease IgY significantly reduced gastric colonization, indicating its preventive efficacy (dose-dependent, up to 500 mg/kg) [128], while clinically, a two-week dietary intake of multivalent anti-*H. pylori* IgY improved symptoms, enhanced quality of life, and increased eradication rates by 30.6% [103]. In a murine model of acute graft-versus-host disease, dietary anti-*E. coli* IgY administration reduced intestinal inflammation and improved outcomes by lowering the expression of inflammatory markers and pathogen-associated receptors [43]. IgY targeting enterotoxigenic *E. coli* (ETEC) reduced bacterial adherence to epithelial cells by 74%, neutralized heat-labile toxin effects, and significantly reduced fluid accumulation in an ileal loop model, confirming both antitoxin and anti-adhesion properties [35]. Oral administration of anti-ETEC IgY in mice further ameliorated clinical signs, restored intestinal integrity, and suppressed proinflammatory cytokine production during enterotoxigenic *E. coli* infection [75]. Similar efficacy was observed for IgY targeting diarrheagenic *E. coli*, which inhibited bacterial growth and adhesion and significantly reduced colonization in an in vivo infection model [84]. In a murine model of urinary tract infection, IgY-rich egg-based formulations significantly reduced the uropathogenic *E. coli* (UPEC) burden over a 15-day treatment period, supporting their potential use as adjuvant therapies [129].

The application of IgY antibodies in the treatment of *Pseudomonas aeruginosa* (PA) infections has yielded mixed results across various experimental models. Several studies have reported encouraging outcomes in murine models of pneumonia and burn wound infections, where passive immunization with polyclonal anti-PA IgY conferred broad-spectrum protection, likely due to its multivalency and strain-independent targeting capability [29]. Anti-flagellin IgY antibodies also showed dose-dependent, non-type-specific protection, achieving full protection in acute pneumonia models and partial efficacy in burn wound infections depending on the flagellin subtype [130]. Immunotherapy with anti-OprF IgY significantly improved survival in PA-infected burned mice and inhibited bacterial invasion in epithelial cell lines, supporting its role in passive protection [131]. Synergistic effects between IgY and antibiotics such as azithromycin, ceftazidime, imipenem, and meropenem were observed both in vitro and in vivo, suggesting that IgY can potentiate conventional antimicrobial therapies and help suppress pulmonary colonization, including in models of cystic fibrosis [132,133]. Furthermore, inhaled co-administration of anti-PA IgY with the macrocyclic peptide POPSICAL led to a >99% reduction in biofilm and bacterial burden in mouse lungs, demonstrating promising antibiofilm activity [134]. However, several studies have reported limited or no protective efficacy under certain conditions. In a rabbit sepsis model, anti-chimeric PilQ/PilA IgY failed to reduce mortality or bacterial burden [135]. Similarly, in a porcine model of ventilator-associated pneumonia, intravenous IgY administration did not significantly reduce pulmonary PA loads, likely reflecting the limited systemic bioavailability of IgY when delivered intravenously [94].

IgY antibodies have also demonstrated antimicrobial potential against a variety of other bacterial pathogens. In a comparative study, anti-*Staphylococcus aureus* IgY produced using Freund’s incomplete adjuvant significantly inhibited bacterial growth, with higher antigen-binding activity compared to other adjuvants [60]. In a murine model of *Salmonella typhimurium* infection, specific IgY antibodies, when delivered via alginate nanoparticles, provided full protection, whereas non-encapsulated IgY was only partially protective, highlighting the value of nanoformulations for enhancing efficacy [136]. Studies on *Vibrio cholerae* showed that anti-*V. cholerae* O1 IgY could significantly reduce mortality in infected mice, with survival rates improving with higher IgY doses [41], while antibodies targeting a chimeric outer membrane protein similarly demonstrated high immunoreactivity and reduced colonization [34]. In the case of *Enterococcus faecalis*, IgY antibodies against cytolysin protected microbiota-humanized mice from ethanol-induced liver damage by neutralizing cytotoxic effects in primary hepatocytes [137].

The use of IgY antibodies in dentistry has emerged as a promising non-invasive strategy for the prevention and management of oral infectious diseases, particularly dental caries, periodontitis, and dentin hypersensitivity. Anti-*Streptococcus mutans* IgY significantly reduced bacterial adhesion and growth in caries-active children following administration via mouth spray or lozenges, indicating its potential as a passive immunotherapeutic in high-risk pediatric populations [138,139]. Similarly, anti-*Porphyromonas gingivalis* IgY inhibited bacterial adhesion and invasion in human periodontal ligament fibroblasts, supporting its use in the prevention and treatment of periodontal disease [42]. Broader-spectrum formulations targeting multiple periodontitis-associated pathogens, such as *Aggregatibacter actinomycetemcomitans* and *Porphyromonas gingivalis*, demonstrated the ability to bind bacterial cell walls, promote agglutination, and disrupt biofilm formation while maintaining stability under varied pH, temperature, and storage conditions [24]. In clinical evaluations, the application of IgY as a local drug delivery agent or mouthwash adjunct to scaling and root planning significantly reduced colony-forming units in both gingivitis and periodontitis patients [140]. Additionally, an IgY–amorphous calcium phosphate (ACP) complex demonstrated dual functionality by occluding dentinal tubules under acid challenge and inhibiting *Streptococcus mutans* biofilms, highlighting its promise in dentin hypersensitivity (DH) management [141].

In the context of fungal infections, IgY antibodies have demonstrated promising antifungal activity against *Candida albicans*, a common opportunistic pathogen. IgY raised against the iron permease Ftr1 from *C. albicans* exhibited high-affinity binding and fungicidal activity, leading to increased survival in *Galleria mellonella* larvae infected with *C. albicans* [32]. In a murine model of vaginal candidiasis, hyperimmune egg-based IgY formulations significantly reduced fungal burden—achieving a 99.6% reduction in colony-forming units (CFUs) by day 15—and diminished inflammation and filamentous yeast forms in vaginal lavage samples, supporting its utility as an adjuvant therapy [142].

The use of IgY has extended beyond human medicine into a wide variety of veterinary applications. In broiler chickens, feed supplementation with microencapsulated IgY significantly improved average daily gain and reduced feed conversion ratio, outperforming conventional antibiotics and supporting more sustainable practices [143]. Anti-*Clostridium perfringens* IgY alleviated clinical signs and normalized hematological and biochemical parameters in experimentally infected birds [107], while maternally or orally delivered IgY targeting *Escherichia coli*, *Salmonella enteritidis*, and *Clostridium perfringens* reduced pathogen colonization, improved immune responses, and was more effective when used prophylactically [105]. Against viral infections, intramuscular administration of anti-Newcastle disease virus IgY improved hemagglutination inhibition titers and survival in broilers, with better performance observed for intramuscular compared to subcutaneous routes [144]. In ducks, specific IgY provided 100% protection against *Riemerella anatipestifer* when administered shortly after infection [145] and markedly reduced mortality from avian influenza H5 virus in Muscovy ducks [83] (Table 2).

IgY-based immunotherapies have also shown promising results in companion animals and livestock. In dogs, chicken-derived IgY-scFv antibodies targeting the capsid protein of canine parvovirus were used for diagnostic and neutralization purposes [147]. In cats, dietary anti-Fel d 1 IgY reduced exposure to the major allergen without affecting feline health or growth [36], and a human proof-of-concept study demonstrated that feeding cats anti-Fel d 1 IgY led to a notable reduction in allergy symptoms among sensitized individuals [148]. In calves, *E. coli* K99-specific IgY, combined with probiotics, improved growth, reduced diarrhea incidence, and supported immune function during the pre-weaning period [104], while IgY targeting bovine coronavirus significantly delayed the onset and reduced the severity of clinical episodes and viral shedding in neonatal diarrhea models, with successful scale-up for industrial production [30]. In equine applications, IgY against *Salmonella enterica* serovar Newport inhibited bacterial motility and growth in vitro in a dose-dependent manner [149]. More broadly, IgY has demonstrated cross-species growth-inhibitory effects against key zoonotic pathogens including *Campylobacter jejuni*, *Salmonella* spp., and *Escherichia coli*, supporting its utility as a sustainable alternative to antibiotics in livestock management [150].

The frequent use of antibiotics in aquaculture to control bacterial diseases has led to the rise of the antimicrobial resistance of the pathogenic bacteria in this sector of activity. As an alternative, IgY antibodies can successfully be used for their protective effects against viruses like nervous necrosis virus (NNV) [151,152] and tilapia lake virus(TiLV) [153] and infections produced by *Vibrio parahaemolyticus* [154], *Aeromonas veronii* [40], *Aeromonas hydrophila* [155], *Edwardsiella tarda* [156], or *Yersinia ruckeri* [157].

IgY-based antivenoms have also been developed, demonstrating the ability to neutralize cobra venom (*Naja naja*) [158,159], krait (*Bungarus caerulens)* [158], white-lipped pit viper (*Trimeresurus albolabris*) [160], crossed pit viper (*Bothrops alternatus*) [161], scorpion (*Tityus caripitensis*) [162], and bee venom (*Apis mellifera*) [163].

Further applications in aquaculture, poultry, and human dental care reveal the broad spectrum of IgY utility, highlighting its role as an alternative to antibiotics. Collectively, these findings support IgY’s incorporation in One Health strategies to combat antimicrobial resistance.

### 4.3. Clinical Trials Using IgY

Clinical trials investigating IgY have shown promising results across diverse medical conditions, including infectious, gastrointestinal, respiratory, and inflammatory diseases. As summarized in Table 3, studies registered on ClinicalTrials.gov highlight both prophylactic and therapeutic applications of IgY. For instance, a completed trial (NCT02705885) evaluated IgY antibodies against *Porphyromonas gingivalis* as a dietary supplement in patients with chronic periodontitis, supporting its role in oral health. Notably, several trials (NCT00633191 and NCT01455675) explored the efficacy of anti-*Pseudomonas* IgY in cystic fibrosis patients, demonstrating its potential in preventing recurrent pulmonary infections. Furthermore, Phase I studies (NCT04567810 and NCT06702280) assessed the safety of IgY against SARS-CoV-2 and its impact on gut microbiota in healthy individuals. Another investigation (NCT02721355) tested anti-*Helicobacter pylori* IgY for chronic gastritis, while NCT03058224 targeted fibromyalgia with IgY formulations against *E. coli* and *Salmonella.* Of the registered clinical trials, only one (NCT02721355) has published results to date reporting a non-serious adverse effect (diarrhea) following the use of an oral IgY-based dietary supplement. These findings underscore IgY’s expanding clinical relevance, with a favorable safety profile and broad-spectrum applicability in human health.

## 5. Advantages and Limitations of Using IgY Compared to IgG

Although chicken egg yolk immunoglobulins are not yet fully utilized, they offer several advantages over their mammalian counterparts. First, IgY technology enhances animal welfare and helps better meet the requirements of the 3R principle: reduction, because laying hens produce a higher yield of immunoglobulins per month (1000–2800 mg/chicken/month) compared to rabbits (200 mg/rabbit/month), thus reducing the number of animals required [17]; replacement and refinement, as antibodies are secreted directly into the egg, eliminating the need for invasive collection procedures, such as blood sampling or sacrifice [65].

IgY technology also possesses economical advantages over conventional antibody production as the cost of maintaining a laying hen is less expensive than maintaining mice or rabbits, while the lower number of animals needed further contributes to the cost-effectiveness of this method [4]. A chicken usually lays around 280 eggs in a year, and with an average egg yolk containing 100–150 mg of IgY, this can result in a yield of 28–42 g of IgY per year per chicken [16]. Although the laying capacity decreases in the second year of egg laying, this is compensated by a greater yield of IgY per egg [168]. Other avian species can be used for IgY production, such as ducks, geese, ostriches, quails, or emu [17,169].

The structural and phylogenetic differences between IgY and IgG result in distinct molecular and biochemical interactions. The Fc region of IgY does not activate the human complement system, bind to rheumatoid factor, or interact with protein G [4]. Avian immunoglobulin does not cross-react with human anti-mouse antibodies (HAMAs), and due to their phylogenetic divergence, avian species elicit a stronger immune response against conserved mammalian proteins [17]. Compared to IgG antibodies produced by mammals, IgY antibodies have similar affinity and avidity [54]. Hens can also generate a strong immune response with lower antigen quantities compared to mammals, thereby reducing the amount of antigen needed for antibody production [16].

In regard to oral IgY therapy, inactivation by digestive processes can be an important challenge as IgY is sensitive to gastric acidity and protease enzymes such as pepsin [170]. In order to overcome this challenge, the use of microencapsulation using liposomes, chitosan–alginate, or whey protein-based microcapsules, and macroencapsulation in enteric-coated gelatin capsules, or even the concurrent use of cytoprotective agent sucralfate has been proposed, with various degrees of success [168]. A summary of these approaches is presented in Table 4.

Another concern relates to the triggering of allergic, systemic, and local responses [171] as IgY may function as an antigen itself. The repeated long-term administration of three IgY formulations in non-microbially challenged mice showed that in the absence of significant amounts of their target antigens, IgYs can influence various immunological processes in ways that are not yet completely understood or easy to predict [172]. In addition, a potential risk associated with the broader use of immunoglobulin-based technologies, including IgY, involves unintended environmental effects, particularly in agriculture and aquaculture where large-scale application is proposed. Nevertheless, a recent study reported that IgY has a significantly higher ecological sustainability value than antibiotics, suggesting a lower likelihood of adverse environmental impact [173].

The lack of a standardized IgY production protocol may be another factor that hampers the widespread use of this technology. Although attempts have been made to standardize different steps of production [6,27], variability exists in immunization schedules, adjuvant selection, IgY extraction, and purification methods, as well as storage conditions, all of which can influence yield and antibody quality. The inherent batch-to-batch variability in polyclonal IgY derived from different hens can also result in inconsistent antibody composition and affinity, necessitating careful validation for reproducibility [174]. To date, the European Centre for the Validation of Alternative Methods (ECVAM) remains the only international body to have issued guidance, through a 1996 report, on IgY production standardization—with no further updates since then.

Several other challenges hinder the clinical translation of IgY. These include limited large-scale clinical trials, underdeveloped regulatory frameworks, and the absence of therapeutic guidelines—especially for human use. While IgY shows promise across both veterinary and human applications, its advancement toward widespread clinical use remains constrained by unresolved questions around immunogenicity, long-term safety, and delivery methods. Additionally, the lack of publicly available clinical trial results further complicates progress as it limits reproducibility and informed development. Addressing these challenges through further research and collaboration is essential to establish consistent protocols and support the broader adoption of IgY technology.

**Table 4 ijms-26-06380-t004:** Comparative summary of encapsulation strategies for oral IgY delivery and their reported outcome.

Reference	Encapsulation Method	Reported Outcome
Lu et al., 2025 [175]	Gel beads with pH-sensitive materials (shellac + lecithin)	Improved IgY stability and enabled controlled release in the small intestine.
Jin et al., 2023 [143]	Microencapsulation (alginate–chitosan)	Improved IgY passage through the gastrointestinal environment.
Li et al., 2022 [176]	Chitosan-coated liposomes (CO_2_-assisted method)	Delayed gastric release and enabled targeted intestinal delivery.
Zhang et al., 2020 [177]	Microencapsulation (alginate–chitooligosaccharide)	Preserved IgY activity during gastrointestinal digestion.
Xing et al., 2017 [178]	Microencapsulation (pectin-based)	Protected IgY in upper GI and enabled delivery to the colon.
Zhou et al., 2010 [179]	Enteric coating + cytoprotective agent (sucralfate)	Improved IgY stability against low pH, pepsin, and freeze–thaw and prolonged storage.
Li et al., 2007 [180]	Microencapsulation (alginate–chitosan)	Improved IgY stability to acid and pepsin.

## 6. Conclusions

Immunoglobulin Y represents a promising immunotechnological platform, offering several advantages over conventional mammalian IgG-based approaches. In the context of rising antimicrobial resistance and the integrative framework of the One Health concept, IgY is increasingly recognized as a valuable tool in diagnostics, prophylaxis, and therapeutics—at the intersection of human health, animal health, and food safety. Its broad applicability, safety profile, and adaptability to scalable production underscore its potential for both antimicrobial resistance management and rapid deployment during future pandemics, where alternative immunotherapies may be urgently needed. The use of IgY is steadily expanding, supported by a growing body of published research and the emergence of commercial products. With continued efforts to address current limitations, standardize production protocols, and further evaluate its safety profile, IgY technology is well positioned to become an accessible and versatile solution for a broad range of applications in the near future.

## Figures and Tables

**Figure 1 ijms-26-06380-f001:**
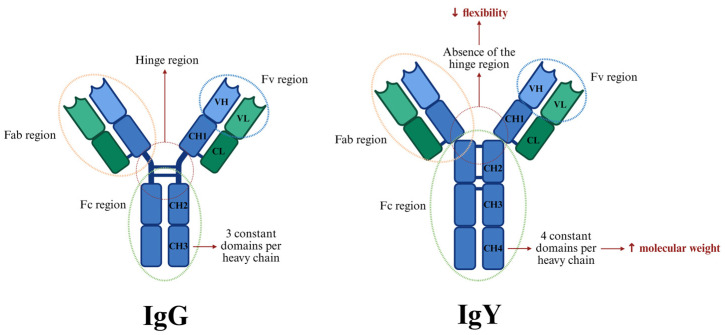
Structural comparison between immunoglobulin G (IgG) and immunoglobulin Y (IgY). Fab—fragment antigen-binding; Fc—fragment crystallizable; Fv—variable fragment; VH—variable region of the heavy chain; VL—variable region of the light chain; CL—constant region of the light chain; CH—constant region of the heavy chain, ↑—increased, ↓—decreased.

**Figure 2 ijms-26-06380-f002:**
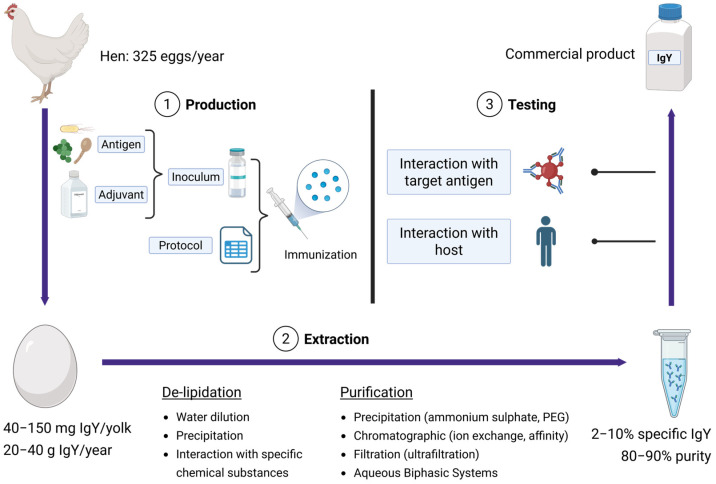
Schematic representation of immunoglobulin Y production technology.

**Figure 3 ijms-26-06380-f003:**
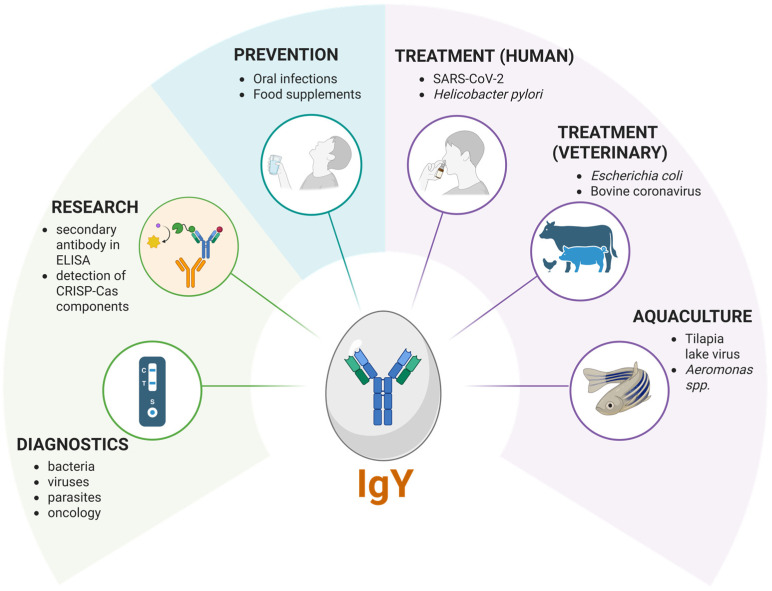
Biomedical applications of IgY: practical use cases across sectors.

**Table 1 ijms-26-06380-t001:** ECVAM recommendation relating to chicken immunization protocols [6].

Factor	Recommendation
Adjuvant	Freund’s incomplete adjuvant, Specol, lipopeptide (Pam_3_-Cys-Ser-[Lys]_4_; 250 µg)
Antigen dose	10 ng–1 mg (preferably 10–100 µg)
Injection site	Intramuscular (field studies; young laboratory chickens) Subcutaneous (older laboratory chickens)
Injection volume	<1 mL
Injection frequency	2–3 times; boosters during laying period
Vaccination interval	4–8 weeks
Use of chickens	Entire laying period (about 1 year)

**Table 2 ijms-26-06380-t002:** Summary of selected articles evaluating the efficacy of IgY-based interventions in poultry.

Reference	Target Pathogen	Route of Administration	Efficacy
Soltani et al., 2025 [106]	*Campylobacter jejuni*	Oral, in diet	Significantly reduced *C. jejuni* counts in cecal and liver tissues
Radwan et al., 2024 [144]	Newcastle disease virus	Subcutaneous or intramuscular	Reduced mortality Improved feed conversion rate
Jin et al., 2023 [143]	Intestinal pathogens	Oral, microencapsulated in diet	Increased average daily gain Improved the feed conversion ratio Reduced colonic *Escherichia coli* and *Salmonella* spp. levels
Qandoos et al., 2023 [105]	*Escherichia coli* *Salmonella enteritidis* *Clostridium perfringens*	Oral	Enhanced performance parameters Reduced bacterial re-isolation
Abadeen et al., 2021 [107]	*Clostridium perfringens* Type A	Oral or intramuscular	Ameliorated the effects of *C. perfringens* type A infection on growth performance, hematology, and serum biochemistry
Yang et al., 2020 [145]	*Riemerella anatipestifer*	Intramuscular	Specific IgY provided 100% curative rate when administered 1 h after infection
Xu et al., 2013 [146]	*Eimeria tenella*	Oral, in diet	Reduced mortality, increased body weight gain (BWG), reduced oocyst shedding, and increased anti-coccidial index

**Table 3 ijms-26-06380-t003:** Clinical trials (ongoing and complete) investigating the use of IgY in human health.

Study Overview	Condition/Goal	Intervention/Treatment	Study Completion
Evaluation of IgY Antibody Effectiveness in Supportive Therapy of Periodontitis Patients [164] Study ID Number: NCT02705885	Chronic Periodontitis	IgY antibodies against *Porphyromonas gingivalis gingipains* as dietary supplement	March 2015
Phase I and II: Post Marketing Study of Anti-pseudomonas IgY in Prevention of Recurrence of *Pseudomonas aeruginosa* Infections Infections in Cystic Fibrosis (CF) Patients [165] Study ID Number: NCT00633191	Cystic fibrosis. Infection with *Pseudomonas aeruginosa*	Anti-pseudomonas IgY gargle	December 2012
Phase III Study to Evaluate Clinical Efficacy and Safety of Avian Polyclonal Anti-Pseudomonas Antibodies (IgY) in Prevention of Recurrence of *Pseudomonas aeruginosa* Infection in Cystic Fibrosis Patients [166] Study ID Number: NCT01455675	Cystic fibrosis	Avian polyclonal anti-pseudomonas antibodies (IgY)	June 2017
Phase I: Assessing the Safety of an IGY Supplement on the Gut Microbiome Study ID Number: NCT06702280	Healthy	Dietary Supplement: IgY supplement	Estimated October 2025
Phase I and II: A Randomized, Double-blind, Placebo-controlled, Parallel Dose Ranging Study on the Influence of IgY Max on Inflammatory Markers and the Gut Microbiome Study ID Number: NCT02972463	Healthy	Immunoglobulin Y	January 2018
A Phase 1 Study in Healthy Participants to Evaluate the Safety, Tolerability, and Pharmacokinetics of Single-Ascending and Multiple Doses of an Anti-Severe Acute Respiratory Syndrome Coronavirus 2 (SARS-CoV-2) Chicken Egg Antibody (IgY) [167] Study ID Number: NCT04567810	COVID-19	anti-SARS-CoV-2 IgY	December 2020
Evaluation of a Health Food Supplement Containing Anti-*Helicobacter pylori* Urease IgY Antibody on Patients with Chronic Gastritis in Hanoi, Vietnam Study ID Number: NCT02721355	Chronic Gastritis Caused by *Helicobacter pylori*	GastimunHP	June 2016
Randomized, Double-blind, Placebo-controlled Exploratory Trial to Investigate Efficacy and Safety of IGN-ES001 in Patients with Chronic Widespread Pain with or Without Fibromyalgia Study ID Number: NCT03058224	Chronic Widespread Pain Fibromyalgia	Polyclonal avian immunoglobulin IgY containing specific IgY against *E. coli* F18ab and *S. typhimurium* in partially delipidated avian egg yolk powder	December 2017

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
