# Peer review of "Latest Findings in Immunoglobulin Y Technologies and Applications"

_ijms, 2025, doi:10.3390/ijms26136380_

Round 1
Reviewer 1 Report
Comments and Suggestions for Authors
Latest findings in immunoglobulin Y technologies and applicationsGeneral Overview
Immunoglobulin Y (IgY) is reviewed in detail in this publication, including its structure, synthesis, medicinal and veterinary uses, and its expanding importance in the One Health paradigm. With its worldwide relevance in diagnostics and antibiotic substitutes in the face of growing antimicrobial resistance, the topic is current. The literature cited is wide and recent, and the paper is generally well-structured.
Detailed Section-by-Section Review
- Title and Abstract
- Line 2–3: The title provides the right amount of information. But, to be consistent with journal style, think about capitalizing the entire title: "Latest Findings in Immunoglobulin Y Technologies and Applications".
- Line 8–24 (Abstract): Concise and well-written, successfully addressing the main themes. To give the abstract a clearer emphasis, it could be helpful to include one or two particular current uses (such as veterinary diagnostics or SARS-CoV-2).
- Introduction
- Lines 27–58: The topic's context is effectively established in the introduction. It explains maternal immunity in detail and highlights the special characteristics of IgY.
- Comment (Line 38): For a more scholarly tone, think about changing "has emerged as a promising tool" to "has gained prominence as a versatile tool
- Comment (Line 46): You could briefly explain how the production volume of IgY differs from that of conventional mammalian antibodies.
- Structure and Function of IgY
- Lines 60–121: This section is easy to understand and has relevant analogies to IgG.
- Comment (Line 64): Reader comprehension would be much improved by a figure that contrasts IgY and IgG side by side with obvious structural differences.
- Comment (Line 90–98): The complement interaction explanation is crucial. Think about elucidating its implications for ELISA kits and clinical testing.
- Comment (Line 103–104): Provide further details about how the improved diagnostic specificity is a result of this more robust immune response.
- IgY Production and Extraction
- Lines 123–283: Very detailed, covering immunization, extraction, and purification.
- Comment (Line 137–140): The reference to the ECVAM protocol is helpful. It would be helpful to include a quick explanation of the three Rs in ethical improvements.
- Comment (Line 198): There is a brief mention of oral route vaccination. If there is research proving its effectiveness or limits, kindly elaborate.
- Comment (Line 245): “Suitable for oral administration” – cite one or two examples where orally administered IgY has succeeded in humans or animals.
- Evaluation and Stability
- Lines 284–308: Technical depth that is appropriate.
Comment (Line 296): Defining the circumstances (such as buffer composition) in which each extinction coefficient is applicable may be useful.
Comment (Line 303): It is said that Western blotting is "commonly used." Kindly provide an example or reference from the clinical studies that are being cited.
- Biomedical Applications of IgY
- Lines 354–593: This section is well-structured and full with evidence.
- Comment (Line 366–367): Although the "Leishmania braziliensis" case is new, readers from other domains will benefit from further background information on CRISPR diagnostics.
- Comment (Line 372–374): Stress the importance of decreased biofilm formation, as this is a crucial discovery for AMR tactics.
- Comment (Line 427–429):
- Comment (Line 439–443): Include more details on the delivery method (nasal spray, dosage, duration) for reproducibility.
Comment (Line 484–487): Provide a summary of the IgY conjugates' relative performance with IgG in the aforementioned assays (ELISA, WB, IF).
- Clinical Trials
- Lines 594–608: This is a solid and properly referenced section.
- Comment (Line 599–602): To give a fair perspective, think about talking about the reasons why some studies (such the one for cystic fibrosis) were stopped or failed, if known.
- Comment (Line 607): Extend the safety profiles mentioned in these trials a little bit; this is important for translational impact.
- Advantages and Limitations
- Lines 612–658: Balanced and informative.
Comment (Line 639–644): The encapsulation strategies for oral delivery are crucial. Suggest adding a diagram or a comparative table for encapsulation methods and their effectiveness.
Comment (Line 650–657): Mention any efforts from WHO or other international bodies to standardize IgY production.
- Conclusion
- Lines 659–669: The conclusion wraps up the article effectively.
Comment (Line 663): Consider emphasizing the translational potential of IgY in future pandemic preparedness and AMR management explicitly.
Minor Editorial Suggestions
- Consistency: Use consistent formatting for abbreviations. Some full names (e.g., ELISA, CRISPR) are repeated without abbreviation.
- Typographical Errors:
- Line 16: “Production and purification strategies are outlined…” → consider breaking into two sentences for clarity.
- Line 253: “The main mechanism being…” – revise to “The primary mechanism is...”
Figures and Tables
- Figure 1 & Figure 2: Visually informative.
Suggestion: Consider including one more figure showing IgY biomedical applications across fields (e.g., diagnostics, oral therapeutics, veterinary use)—perhaps as a flowchart or infographic.
Overall Recommendation
Recommendation: Minor Revision
With a few minor edits, this work is ready for publishing and is quite informative. The writers deserve praise for their thorough analysis of the literature, well-organized presentation, and lucid explanation of intricate immunological ideas. Resolving the aforementioned issues will improve the paper's readability, scientific soundness, and clarity.
Reviewer 2 Report
Comments and Suggestions for Authors
Dear authors
I hope this finds you all well. Regarding the review of manuscript number IJMS- 3641487, entitled "Latest findings in immunoglobulin Y technologies and applications". It is indeed an interesting review. Minor revision should be carried out.
Comments
- Line 148-217: All these points can be categorized under the title of factors optimizing IgY production.
- Please highlight the pharmacokinetic and pharmacodynamic profiles of IgY in lines 317-352.
- Mention, in a separate table or merge with table 2, the previous studies of IgY therapy facing poultry diseases and add a column in the same table indicating the route of administration and efficacy.
Reviewer 3 Report
Comments and Suggestions for Authors
This is a comprehensive, well-organized review focusing on the structure, production, extraction, and broad applications of Immunoglobulin Y (IgY), with special emphasis on its role in diagnostics, therapeutics, and prophylactics in human and veterinary medicine. The manuscript covers recent advances (mainly within the last five years) and discusses current limitations and future prospects of IgY technology. Overall, this is a timely and relevant topic given the increasing interest in alternatives to mammalian antibodies and antibiotic resistance management.
- While the review is descriptive and comprehensive, it would benefit from more critical analysis. Specifically: What are the key challenges preventing large-scale clinical translation? What are the controversies (if any) in IgY applications or limitations in veterinary vs. human uses?
- The safety profile of IgY is highlighted, but potential risks such as allergic responses, immune interference, or unintended environmental effects (especially in aquaculture or agriculture) should be acknowledged more explicitly.
- The manuscript is very long, with some sections (e.g., production and extraction protocols) containing repetitive explanations or too much procedural detail. Consider condensing technical descriptions or moving less critical procedural content into a table or figure.
